# Intraspecific Variability in *Trionymus aberrans* Goux, 1938 (Hemiptera: Coccomorpha: Pseudococcidae) with Description of the Second-Instar Nymph

**DOI:** 10.3390/insects14120945

**Published:** 2023-12-13

**Authors:** Małgorzata Kalandyk-Kołodziejczyk, Marcin Walczak, Elżbieta Podsiadło, Barbara Franielczyk-Pietyra, Jolanta Brożek

**Affiliations:** 1Faculty of Natural Sciences, Institute of Biology, Biotechnology and Environmental Protection (IBBEP), University of Silesia in Katowice, Bankowa 9, 40-007 Katowice, Poland; malgorzata.kalandyk@us.edu.pl (M.K.-K.); marcin.walczak@us.edu.pl (M.W.); barbara.franielczyk-pietyra@us.edu.pl (B.F.-P.); 2Department of Animal Environment Biology, Warsaw University of Life Sciences SGGW, Ciszewskiego 8, 02-786 Warszawa, Poland

**Keywords:** nymphal stages, adult females, morphology, spoil heaps, scanning electron microscopy

## Abstract

**Simple Summary:**

*Trionymus aberrans* Goux, 1938 (Hemiptera: Coccomorpha: Pseudococcidae) lives inside the leaf sheaths of grasses. Adult females collected in post-industrial wastelands differ morphologically from those found in undegraded habitats. We observed unusual structures in specimens from contaminated areas. These structures, never observed before in members of the Pseudococcidae, were studied using scanning electron microscopy. A detailed description of the second-instar nymph is provided, which will facilitate the identification of this stage.

**Abstract:**

The morphological characteristics of adult females of *Trionymus aberrans* Goux, 1938 (Hemiptera: Coccomorpha: Pseudococcidae) collected from post-industrial wastelands and other habitats in Poland and other countries were studied. Distinctive morphological variations were observed in the specimens from post-industrial wastelands in Poland. Scanning electron micrographs of the morphological characters of *T. aberrans* are provided. The presence of unusual pores, each with four loculi, was demonstrated for the first time in a species of Pseudococcidae. The importance of introducing additional morphological characters into the species description is discussed. New data on the frequency and host preferences of *T. aberrans* are also provided. This research is the first long-term study on scale insect species in post-industrial wastelands. The second-instar nymph of *T. aberrans* is described and illustrated and the presence of translucent pores on the hind coxae of this developmental stage is reported for the first time.

## 1. Introduction

The Pseudococcidae (mealybugs) is subdivided into three subfamilies—the Phenacoccinae, Pseudococcinae [1,2,3,4,5] and Rastrococcinae [6] and it comprises the second largest family within the scale insects (Hemiptera: Coccomorpha) with 2049 species [7].

The genus *Trionymus* Berg, 1899, which comprises 121 species, is one of the largest genera of the subfamily Pseudococcinae [7]. Adult female *Trionymus* are characterized by: the presence of 6–8-segmented antennae; legs normally developed; claws without denticles; both pairs of dorsal ostioles frequently well developed; swirled trilocular pores present on both body surfaces; multilocular disc-pores always present on the venter of the abdomen, and frequently on its dorsum; tubular ducts on both body surfaces; cerarii reduced in number, always including at least the anal lobe pair; abdominal cerarii each with two conical setae; and circulus present or absent [3,8,9]. The genus is distributed worldwide apart from Antarctica. According to Kosztarab and Kozár [9] and other authors [1,3,5], *Trionymus* comprises a number of morphologically similar species and as the species are inadequately described, the genus needs thorough revision.

In their comprehensive review of the Palaearctic mealybug fauna, Danzig and Gavrilov-Zimin [3] distinguished a *Trionymus*-group of genera comprising: *Trionymus* Berg, 1899; *Dysmicoccus* Ferris, 1950; *Balanococcus* Williams, 1962; *Kiritshenkella* Borchsenius, 1948 and *Saccharicoccus* Ferris, 1950.

Only eight species of *Trionymus* occur in Poland, namely: *T. aberrans* Goux, 1938; *T. hamberdi* (Borchsenius,1949); *T. newsteadi* (Green, 1917); *T. perrisii* (Signoret, 1875); *T. phalaridis* (Green, 1925); *T. placatus* (Borchsenius, 1949); *T. radicum* (Danzig, 1986) and *T. thulensis* Green, 1931 [10]. 

*Trionymus aberrans* is distributed in almost all of southern Europe and extends northward into Germany and Poland and eastward into Ukraine and Georgia. The species is also found in Southwest and Central Asia and as far east as East Asia [7]. It occurs inside the leaf sheaths of various grass species [7,11].

Descriptions and illustrations of the adult female *Trionymus aberrans* have been provided by Goux [12], Schmutterer [13], Kosztarab and Kozár [9] and Danzig and Gavrilov-Zimin [3]. 

Preliminary studies revealed an intraspecific variation in specimens of *T. aberrans* found in spoil heaps in Poland [14]. These observations were confirmed during the present research and, moreover, additional features of this species were found.

The structure and arrangement of the wax gland openings (pores and ducts) are the main taxonomic characters used in the classification of all scale insects, including mealybugs. The following types of pores occur in mealybugs: simple discoidal pores, swirled trilocular pores, quinquelocular disc-pores and multilocular disc-pores [2,8,15]. The presence of quadrilocular pores (each with four secretory openings (loculi)) has never been demonstrated in any species of Pseudococcidae before.

Translucent pores are structures found in many mealybug species. The function of these pores is unclear, but possibly they release sex pheromones since they occur almost solely in adult females [15].

The classification and species identification of scale insects is based almost entirely on the morphology of adult females. In the Pseudococcidae, the female has three nymphal stages, whereas the male has four [15]. The morphology of the nymphs of most species of Palearctic Pseudococcidae is undocumented [3]. Only the first-instar nymph of *T. aberrans* was briefly described by Goux [11], but it was not illustrated; its other nymphal stages are not known [3,9]. 

The aim of this study was to compare the morphology and to highlight possible morphological differences between females of *T. aberrans* collected in post-industrial wastelands and those found in undegraded habitats. Moreover, the collection of some *T. aberrans* second-instar nymphs presented the opportunity to describe the morphology of this stage, which is so far unknown.

## 2. Materials and Methods

Adult females of *T. aberrans* were mainly collected from degraded post-industrial wastelands, including three zinc spoil heaps located in Ruda Slaska (site. 1), Piekary Slaskie (site 2) and Boleslaw (site 3), one coal heap in Ruda Slaska (site 4) and a degraded area near a zinc production plant in Piekary Slaskie (site 5) between 2011 and 2021 (Figure 1). Ruda Slaska and Piekary Slaskie are towns, while Boleslaw is a small village located near towns Bukowno and Olkusz. The sites 1, 2, 3 and 5 are heavily contaminated with zinc [16,17,18,19].

All the sites are in southern Poland. Specimens were also collected from undegraded habitats (unnumbered sites), primarily in psammophilous and xerothermic grasslands in the same localities reported above, plus in Austria and the Czech Republic. The post-industrial wastelands were visited regularly each year between April and October, while the unnumbered sites were visited frequently, but not regularly. Adult females were collected by hand from the leaves or leaf sheaths of various grass species and were subsequently preserved in 70% ethanol and then stained with acid fuchsin and slide mounted. The material was identified using the keys in Borchsenius [20], Kosztarab and Kozár [9] and Danzig and Gavrilov-Zimin [3].

Type material and other specimens of *T. aberrans* deposited in the Muséum national d’Histoire naturelle (MNHN), Paris were examined.

*Trionymus aberrans* specimens were identified using a Nikon Eclipse E-600 biological microscope (Boston Industries, Boston, MA, USA) with a Nikon DS-Fi2 digital camera and NIS Elements software (version 4.2, Nikon, Japan) at the Institute of Biology, Biotechnology and Environemental Protection, University of Silesia in Katowice, Bankowa 9, 40–007 Katowice (IBBEP). The females were examined under a dissection microscope before they were slide mounted and the diagnostic character used to identify the species was the presence of a horseshoe-shaped anal ring, a unique character of *T. aberrans*.

For scanning electron microscopy, the method of Kanturski et al. [21] was adapted and used to dehydrate and dry the specimens. The samples were then critical-point-dried in a Leica EM CPD300 automated critical point dryer (Leica Microsystems, Vienna, Austria) mounted on aluminum stubs with double-sided adhesive carbon tabs and sputter coated with gold (20–30 nm thick film) in a Pelco SC-6 sputter coater (Ted Pella, Inc., Redding, CA, USA). Images were captured using a Hitachi SU8010 field emission scanning electron microscope (SEM) (Hitachi, Hitachi High-Technologies Corporation, Tokyo, Japan) at 5.0, 7.0 and 10.0 kV accelerating voltages. The description of the second-instar nymph of *T. aberrans* is based on three specimens. The specimens of *T. aberrans* that were collected during the present study are deposited at IBBEP. Measurements are in micrometers (μm); body length and width were measured in millimeters (mm).

A total of 230 females of *T. aberrans* were examined: 5 from MNHN (including types), 75 were collected from post-industrial wastelands (15 from each site) and 150 from undegraded habitats—including specimens collected from psammophilous grasslands located in Piekary Slaskie, Boleslaw and Ruda Slaska.

## 3. Results

### General Information

A total of about 900 adult female *T. aberrans* were collected, most (620) from the post-industrial wastelands. About 40% of all of the females collected were parasitized. The highest level of parasitization was observed in females collected between August and October.

The females were found on the leaves or in the leaf sheaths of various Poaceae. In the post-industrial wastelands, about 90% of the specimens were collected from *Deschampsia caespitosa* (L.) Palisot de Beauvois. Other grass species, e.g., *Agrostis capillaris* L., *Elymus repens* (L.) Gould, *Phleum pratense* L. and *Poa compressa* L., were also common hosts in degraded habitats. Adult females formed colonies of 4–10 specimens only on *Deschampsia caespitosa* in the post-industrial wastelands. In the undegraded habitats, e.g., psammophilous or xerothermic grasslands, adult female *T. aberrans* were often collected from *Brachypodium pinnatum* (L.) Palisot de Beauvois and *Phleum* sp., and females most often occurred singly on their host plants.

First-instar nymphs of *T. aberrans* were not collected. Second-instar nymphs were collected from both undegraded areas and post-industrial wastelands, but the vast majority of these specimens were highly parasitized. Third-instar females were also collected, but they were not studied because the specimens were significantly damaged. Adult males and male immature instars (prepupa and pupa) were not collected.

Intraspecific variability in adult females of *Trionymus aberrans*

Material examined

Type material:

*Trionymus aberrans* n. sp./Hem. Coccidae/♀/Avena/Marseille/1.VIII.1936/806 /a/Type/Holotype (MNHN); *Trionymus aberrans ovalis*/Type/Colmars (Basses-Alpes)/L. Goux coll/19/VII/1939/ 1013 (MNHN). Three additional adult females are deposited in MNHN, while 225 adult females collected during the present study have been deposited in IBBEP.

Out of the ca. 900 *T. aberrans* adult females collected, 225 were involved in the morphological examination in the present study (Appendix A).

Diagnosis of adult female having the following combination of features: (i) horseshoe-shaped anal ring; (ii) presence of simple tubular ducts of two types (dorsal ducts larger than those of venter); (iii) multilocular disc-pores present on both body surfaces; (iv) presence of two pairs of cerarii; (v) each anal lobe cerarius (C18) containing two conical setae, 6–16 swirled trilocular pores and three or four auxiliary hair-like setae; (vi) circulus usually absent but rarely present; and (vii) antennae each eight (or rarely nine) segmented.

Description of a slide-mounted adult female (based on all studied females): Body elongate oval, 4.25–4.92 mm long, 2.74–3.64 mm wide. Eyes on margin, each 31–32 μm wide. Antennae each eight (rarely nine) segmented, 560–610 μm long; segment lengths (in μm): first segment, 58.13–64.20; second, 50.05–55.8; third, 36.42–41.13; fourth, 26.58–36.77; fifth, 30.68–34.16; sixth, 27.40–31.03; seventh, 31.84–38.03 and apical segment 81.77–93.18. Number of sensory setae on antennal segments: first segment, three; second, four; third, four; fourth, three; fifth, five; sixth, 4–5; seventh, four and apical segment, 16 or 17. Setal lengths (in μm): apical seta 30.71–32.1, subapical seta 24.17–26.15. Clypeolabral shield 180.22–196.61 μm long, 143.63–160.77 μm wide. Labium three-segmented, 74.25–82.32 μm long, 79.78–80.75 μm wide. Stylet loop reaching to level of anterior spiracles. Anterior spiracles each 55.52–65.39 μm long and 28.90–32.58 μm wide across atrium, with three or four associated trilocular pores and one multilocular disc-pore located nearby; posterior spiracles each 67.27–78.35 μm long and 42.53–44.23 μm wide across atrium, with five or six associated trilocular pores and 3–4 multilocular disc-pores located nearby. Circulus usually absent, rarely with one small oval circulus present. Legs well developed; hind leg measurements (in μm): trochanter + femur 230.55–239.72; hind tibia + tarsus 266.15–283.66; hind claw 27.81–29.06. Hind coxae each with translucent pores. Tarsal digitules setose, subequal in length, each 46.45–48.33 μm long. Claw digitules subequal in length, each 27.29–28.46 μm long, both capitate and thicker than tarsal digitules. Both pairs of ostioles present; each anterior ostiole with 4–8 trilocular pores and one or two setae (total for both lips), and each posterior ostiole with 8–12 trilocular pores and two setae (total for both lips). Anal ring horseshoe-shaped 90–110 μm wide, bearing six anal-ring setae, each seta 76.87–91.22 μm long. 

Venter. Body setae slender, each 52.8–61.14 μm long, with the longest setae present medially on the head; apical seta on each anal lobe is 147.95–181.34 μm long. Multilocular disc-pores each 6–8 μm wide, numerous on the abdomen, forming transverse rows and bands across last four abdominal segments, with 1–3 on the head, and present irregularly along body margins. Trilocular pores each 4–5 μm in diameter, scattered over the entire surface. In some specimens, unusual four-locular pores are present on the last three abdominal segments. Simple tubular ducts are of two sizes: larger tubular ducts, each 10.4–12.2 μm long and 3.94–4.71 μm wide at orifice, forming transverse bands across the last four abdominal segments, a group on each margin of abdominal segment VI, also occasionally present on cephalothorax; smaller tubular ducts, each 4.95–6.56 μm long and 2.62–3.1 μm wide, scattered over entire surface.

Dorsum. Derm membranous, with two pairs of cerarii (C17 and C18) on penultimate and last abdominal segments; C17 each with two cerarian setae, 3–6 trilocular pores between cerarian setae, and one or two auxiliary setae; anal lobe cerarii (C18), each with two conical setae, 6–16 trilocular pores and three or four auxiliary hair-like setae. Dorsal body setae hair-like, each 25.2–32.46 μm long, scattered on head and thorax, forming rows across abdominal segments. Multilocular disc-pores each 6.68–8.22 μm wide, forming transverse rows across last three abdominal tergites. Trilocular pores each 3.5–4.25 μm in diameter, scattered over entire surface. Simple tubular ducts of two sizes are scattered throughout body surface, including head and thorax. Tubular ducts of two sizes, slightly larger than ventral ducts: the smaller ducts were each 10.7–12.8 μm long and 3.94–4.9 μm wide at orifice; the larger were 13.5–14.97 μm long and 4.14–5.2 μm wide at orifice.

Comments:

The holotype of *T. aberrans* (MNHN) has the following characteristic features: horseshoe-shaped anal ring; hind coxae with many translucent pores; simple tubular ducts of two types on both body surfaces (dorsal ducts larger than those of venter), numerous swirled trilocular pores on dorsal and ventral body surfaces; multilocular disc-pores present on both body surfaces, two pairs of cerarii, each penultimate cerarius (C17) with two conical setae, two trilocular pores and one auxiliary seta, each anal lobe cerarius (C18) with two conical setae, eight trilocular pores and four auxiliary hair-like setae; circulus absent; and antennae each eight-segmented. 

Morphological intraspecific variation in adult females of *T. aberrans* was observed. The vast majority of females had eight-segmented antennae, but 11 of the collected specimens had nine-segmented antennae; the presence of eight- or nine-segmented antennae apparently was not associated with the habitat type (Table 1).

Swirled trilocular pores were present on both body surfaces in all specimens (Figure 2a,c). Multilocular disc-pores were observed on both dorsal and ventral abdominal segment surfaces. These structures did not show variation among individuals.

Morphological differences in *T. aberrans* were not observed in specimens collected from undegraded psammophilous grasslands in Ruda Slaska, Piekary Slaskie and Boleslaw. No morphological variation in the structure or sizes of simple tubular ducts were observed; two sizes were present in all of the examined specimens.

Differences in number of trilocular pores in C18 were observed in females collected from the four zinc-contaminated post-industrial wasteland sample sites: zinc heaps (sites 1, 2 and 3) and the degraded area near the zinc production plant at Piekary Slaskie (site 5).

Unusual pores were observed in a few females from only one of the post-industrial wastelands sites (Table 1), the degraded area near the zinc production plant in Piekary Slaskie (site 5) collected in 2017, 2018 and 2020. There were 6 to 10 unusual pores in each specimen. These pores each had four secretory openings (loculi) (Figure 2a,b) and were scattered over the venter of the last three abdominal segments. Each of the pores were similar to a normal swirled trilocular pore but had one more secretory loculus, and so were quadrilocular. No pores with four loculi were observed in females that had been collected from undegraded habitats or in any of the females deposited in the collections at MNHN and IBBEP.

A small circulus was rarely observed in females collected during the present study (Table 1).

Translucent pores were present on the hind coxae in all the adult females, but there were differences in the number of these pores between females collected from post-industrial wastelands and those from undegraded habitats (Table 1).

All of the adult females studied had two pairs of abdominal cerarii (C17 and C18), but there were differences found in the morphology of the two pairs of cerarii (Table 1). 

The anal lobe cerarius (C18) most often contained two conical setae, two auxiliary hair-like setae and 6–16 trilocular pores. The number of trilocular pores in C18 differed between females living in different habitats. In some specimens, there was an additional (third) conical seta in C18 (Table 1).

Description of a second-instar nymph of *Trionymus aberrans*

Material examined:

2261/*Trionymus aberrans*/♀/larva/Graminae vagina/det. Koteja//5.9.1967/6/Las Wolski Kraków/leg. Koteja//SC28-249-1-004-DZUS; *Trionymus aberrans*/nymph/*Phleum* sp./Olsztyn near Częstochowa/12.09.2018//leg. et det. M. Kalandyk-Kołodziejczyk; *Trionymus aberrans*/nymph/Deschampsia caespitosa/Piekary Slaskie, Lotników/site 5/12.09.2020/leg. et det. M. Kalandyk-Kołodziejczyk.

Diagnosis of second-instar nymph having the following combination of features: (i) horseshoe-shaped anal ring; (ii) the presence of a few simple tubular ducts of one type only, on the last segments of the abdomen (dorsal ducts larger than those on venter); (iii) the absence of multilocular disc-pores; (iv) the presence of two pairs of cerarii; (v) each anal lobe cerarius (C18) with two conical setae, five swirled trilocular pores and two auxiliary hair-like setae; (vi) circulus absent and (vii) antennae each six segmented.

Description of slide-mounted second-instar nymph (Figure 3) based on three specimens—body elongate oval, 1.1–1.3 mm long, 0.4–0.5 mm wide.

Eyes present on margin, each about 20.18 μm wide. Antennae each six segmented; segment lengths (in μm): first segment, 33.94–38.37; second, 35.40–36.48; third, 39.83–41.80; fourth, 21.17–21.80; fifth, 25.16–25.68 and apical segment, 70.51–71.88. Number of sensory setae on antennal segments: first segment, three; second, three; third, three; fourth, three; fifth, three and apical segment, 13. Apical seta, 25.58–26.21 μm long; subapical seta, 14.11–15.45 μm long. Clypeolabral shield was 137.94–142.62 μm long and 99.53–100.79 μm wide. Labium was 54.09–54.82 μm long and 57.51–56.58 μm wide. Stylet loop reaching to the level of anterior spiracles. Anterior spiracles were each 44.12–44.43 μm long, 16.32–18 μm wide across atrium; posterior spiracles were each 44.12–45.8 μm long, 20.18–21.5 μm wide across atrium. Both anterior and posterior spiracles were each associated with two trilocular pores. Circulus was absent. Legs were well developed: hind leg measurements (in μm): hind trochanter + femur, 148.11–151.65; hind tibia + tarsus, 167.04–168.30; hind claw, 14.74–16.80. Hind coxae each had a few translucent pores. Claw was without denticle. Tarsal digitules were slightly capitate, subequal in length, each 30.87–31.64 μm long. Claw digitules were clubbed, subequal in length, each about 20.16 μm long, and slightly thicker than tarsal digitules. Both pairs of ostioles were present; each anterior ostiole with four trilocular pores and one seta (total for both lips); each posterior ostiole with eight trilocular pores and two setae (total for both lips). Anal ring was horseshoe-shaped, 42.46 μm long and 43.88 μm wide, bearing six anal-ring setae, each seta 64.05–67.63 μm long.

Venter. Body setae slender was hair-like, each 21.53–25.96 μm long, with the longest setae present medially on the head; apical seta on each anal lobe was 107.15–109.14 μm long. Multilocular disc-pores absent. Swirled trilocular pores each scattered evenly over entire surface. Simple tubular ducts were of one type only, very few on lateral margin of abdominal segment VII, each duct 4.87¬5.00 μm long and 3.65–3.85 μm wide at orifice.

Dorsum. Derm membranous, with two pairs of cerarii on body margin; C17 each with one conical seta, two trilocular pores and two hair-like setae; anal lobe cerarii (C18) each with two conical setae, each seta was 8.56–8.82 μm long, five trilocular pores and two auxiliary hair-like setae. Body setae hair-like, shorter than ventral setae, each 13.6–16.24 μm long. Swirled trilocular pores evenly scattered. Simple tubular ducts of one type, and few, were present only on abdominal segments VI–VIII; segments VI and VII had very few ducts on each lateral margin; segment VIII had a few ducts on lateral margins and in middle; each duct was longer than a ventral duct, 5.82–5.88 μm long and 3.5–3.66 μm wide at orifice.

Comments: 

The species identification of second-instar nymphs was based on the presence of a horseshoe-shaped anal ring. According to Gullan et al. [22], the sex of the second-instar nymph can only be determined for specimens that contain a pharate third-instar male or female. However, several authors [23,24,25] report that second-instar male nymphs have more dorsal tubular ducts than second-instar females. 

Translucent pores were observed on the hind coxae of all three of second-instar nymphs of *T. aberrans* examined. Previously, these pores have been recorded almost exclusively in adult females [15].

One of the examined nymphs had been collected from a post-industrial wasteland at Piekary Slaskie (site 5). Further studies with more specimens are necessary to investigate the existence of the intraspecific variability in *T. aberrans* nymphs collected from post-industrial wastelands.

## 4. Discussion

Morphological variability in many species of Pseudococcidae has often been observed, e.g., [2,3,14,25,26,27,28]. As was emphasized by Williams [15], the morphological variations in a single species can be quite striking. A considerable number of morphological variations in some mealybug species appear to be environmentally induced [25,27,29,30]. Mróz et al. [14] observed fewer trilocular pores in the anal lobe cerarii (C18) and a reduction in the number of translucent pores on the hind coxae in specimens of *T. aberrans* found in post-industrial wastelands in comparison to specimens collected in other habitats, but they did not provide details of these morphological features from the five specimens collected from spoil heaps that they examined [14]. 

We observed unusual pores with four secretory openings (loculi) in seven out of fifteen specimens of *T. aberrans* collected from the zinc-contaminated degraded area at Piekary Slaskie (site 5). To the best of our knowledge, the presence of such unusual pores in species of *Trionymus* has never been recorded before, nor has this type of pore ever been described from any other species of Pseudococcidae [3]. The structure of the unusual pores in *T. aberrans* is different from that of the quadrilocular pores found in species belonging to other scale insect families, although they all have four loculi each. We called them “unusual pores with 4 loculi”; their swirled structure suggests that perhaps they are derived from swirled trilocular pores. The presence of these unusual pores may confer some advantageous adaptation to the environmental conditions in a human-transformed and polluted habitat.

Although there are always two pairs of cerarii in adult female *T. aberrans*, our studies revealed that their morphology is variable in different environmental conditions. 

Differences in other features were only observed in adult female *T. aberrans* that had been found in post-industrial wastelands. In our opinion, the larger range of numbers of trilocular pores in the anal lobe cerarii (C18), i.e., from six to sixteen, could be included in the species description. Moreover, four of the females that had been collected from the post-industrial wastelands had an additional thick conical seta in C18, which has not been observed by other authors, e.g., [3,9,20,31]. Further research needs to be conducted to determine whether these features are characteristic of individuals from degraded areas. 

Specimens from the examined post-industrial wastelands differed in their morphological anomalies. Unusual four-locular pores and a circulus were present only in females collected from the degraded area near the zinc production plant in Piekary Slaskie (site 5). Smaller numbers of trilocular pores in C18 were observed in most females collected from this site and from the three zinc heaps (sites 1–3). However, females collected from a coal spoil heap at Ruda Slaska did not possess morphological anomalies except for having fewer translucent pores on the hind coxae. This coal spoil heap was not contaminated with zinc and had been recultivated, so it was covered entirely with vegetation; the environmental conditions were therefore very different from those in the four zinc-contaminated, post-industrial wastelands sampled. In contrast, it is worth emphasizing that the specimens collected from undegraded habitats in the same towns, where the post-industrial wastelands are situated, did not possess morphological anomalies.

Very few researchers have conducted studies on scale insects from post-industrial wastelands other than the works of Lubiarz and Golan [32], Lubiarz and Cichocka [33], Kalandyk-Kołodziejczyk et al. [34] and Mróz et al. [14]. These studies included a few species of the Coccidae in degraded areas [32,33] and scale insect species diversity in post-industrial wastelands [14,34]. The present work is the first long-term study on scale insects found in post-industrial wastelands.

Differences in abundance and host-plant preferences have been observed between the adult females of *T. aberrans* from post-industrial wastelands and those living in undegraded habitats in Poland and other countries. *Deschampsia caespitosa* was a common species in all degraded wastelands and many undegraded areas, but we found females of *Trionymus aberrans* on this grass only in post-industrial habitats. On the other hand, *Brachypodium pinnatum* was present in two types of habitats, but *T. aberrans* were found on this grass only in undegraded habitats. In habitats such as xerothermic and psammophilous grasslands, usually only one or two adult females have been found on a single grass plant [35,36], while in the degraded areas examined in this study, between four and ten females were found on a single plant. 

A significantly higher number of individuals per host plant in heavily degraded areas compared to natural habitats has also been observed in the soft-scale insect *Parthenolecanium rufulum* (Cockerell 1903) (Coccomorpha: Coccidae) [32,33]. The number of individuals of *P. rufulum* found on oaks growing in degraded areas near a nitrogen fertilizer factory in Puławy, Poland were more than 60 times greater than those found in natural habitats in Polesie National Park [33]. This may indicate that difficult environmental conditions, including soil contamination and low humidity in post-industrial areas, are somehow favorable for the reproductive success of some scale insect species, including *P. rufulum* and *T. aberrans*. 

Adult female *T. aberrans* were found in both post-industrial wastelands and undegraded habitats in Poland and other countries between June and October, which is consistent with the results obtained by Schmutterer [13] in Germany. Many of the specimens collected in the present study had been parasitized by a species of Hymenoptera, which Schmutterer [13] also observed. 

*Trionymus aberrans* has hind coxae with translucent pores [3,9,20]. Although these pores were observed in all of the studied material, we found differences in their number between specimens collected from different environments. Females from undegraded habitats generally had numerous translucent pores, whereas those collected from post-industrial habitats had only a few; this had been recorded previously by Mróz et al. [14]. According to Williams [15], the function of these pores is still not clear, but because they are only present in adult females, they probably emit sex pheromones; however, we also observed these pores in a second-instar nymph, so perhaps their function is different. 

Adult females of *T. aberrans* usually do not have a circulus [3,9,20], but a few specimens have one [31,37]. Only five female *T. aberrans* from one post-industrial wasteland (site 5) each had a small circulus between abdominal segments III and IV. According to Danzig and Gavrilov-Zimin [2,3], mealybugs can have from one to five circuli situated on the abdominal venter or may lack a circulus, although the most common condition is a single circulus. In species that generally lack a circulus, one can be present in occasional specimens [15] and females both with and without a circulus can be found together in the same population [3], as shown in our material.

Our research recorded significant morphological variability in adult females of *T. aberrans* in specimens from post-industrial wastelands. Further studies are needed to investigate such morphological variability in other species of Pseudococcidae living in highly contaminated habitats to clarify whether such morphological variability is a characteristic of post-industrial habitats.

The classification and species identification of scale insects is based almost entirely on the morphology of adult females [3,9]. Often, morphological structures such as circuli, conical setae and ostioles are present in the nymphal stages but are absent in the adults and vice versa. Nymphal instars have not been sufficiently studied, despite their having conserved morphological features that can help to resolve evolutionary relationships [24]. Identification to the species level is often impossible for the immature stages and males, although immature stages often make up a large proportion of a sample [38]. Little information is available that enables the species identification of immature instars of the pseudococcids [24]. We provide a description of a second-instar nymph of *T. aberrans*, which is probably an immature female instar.

It is noteworthy that translucent pores were observed on the hind coxae of all the *T. aberrans* second-instar nymphs examined. These pores have been observed almost exclusively in adult females. According to Williams [15], only the Australian species *Sphaerococcus casuarinae* Maskell, 1892 has similar pores in its intermediate instars and there is a great need for taxonomic revisions of mealybugs that are based on the external morphological characters of the different instars as well as the adult females.

## Figures and Tables

**Figure 1 insects-14-00945-f001:**
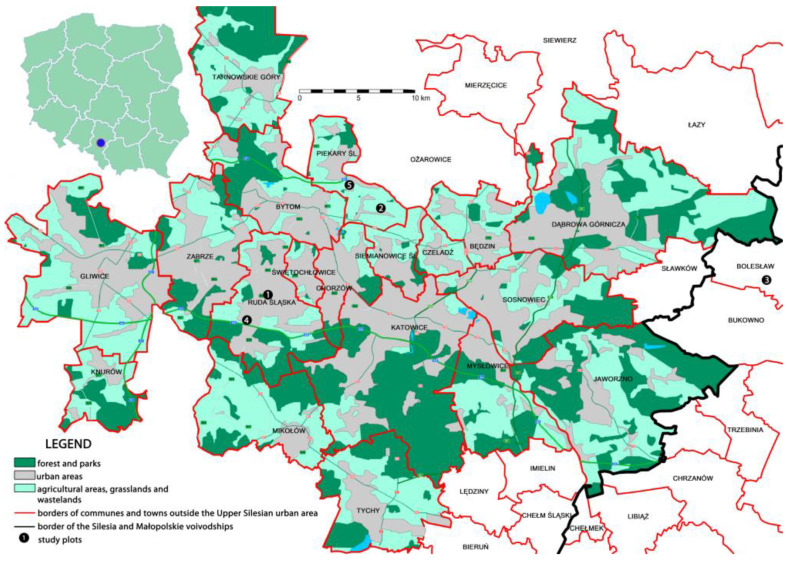
Sampling sites (post-industrial wastelands) in Upper Silesia, Poland. Map from https://pl.wikipedia.org/wiki/Konurbacja_górnośląska/media/Plik:Konurbacja_górnośląska_19_GUS_2013.png CC BY-SA 3.0, accessed on 20 July 2023, modified with Adobe Photoshop 8.0.

**Figure 2 insects-14-00945-f002:**
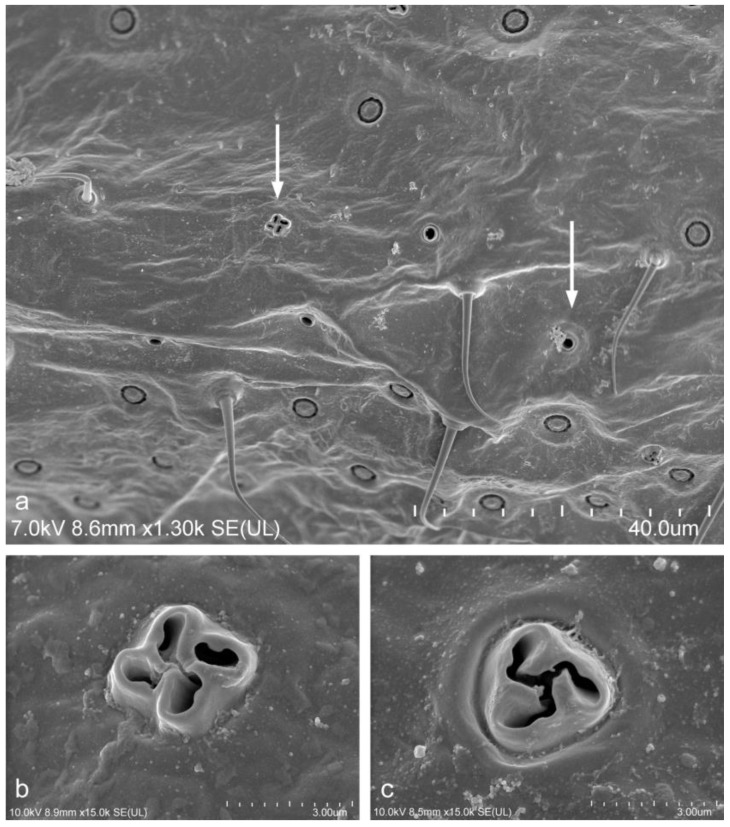
Abdominal segment of adult female of *Trionymus aberrans* Goux, 1938 (SEM) collected from a post-industrial wasteland in Poland. (**a**) Ventral view of an unusual pore with 4 loculi and simple tubular ducts (each marked with a white arrow); (**b**) normal pore with 4 loculi; (**c**) usual swirled trilocular pore.

**Figure 3 insects-14-00945-f003:**
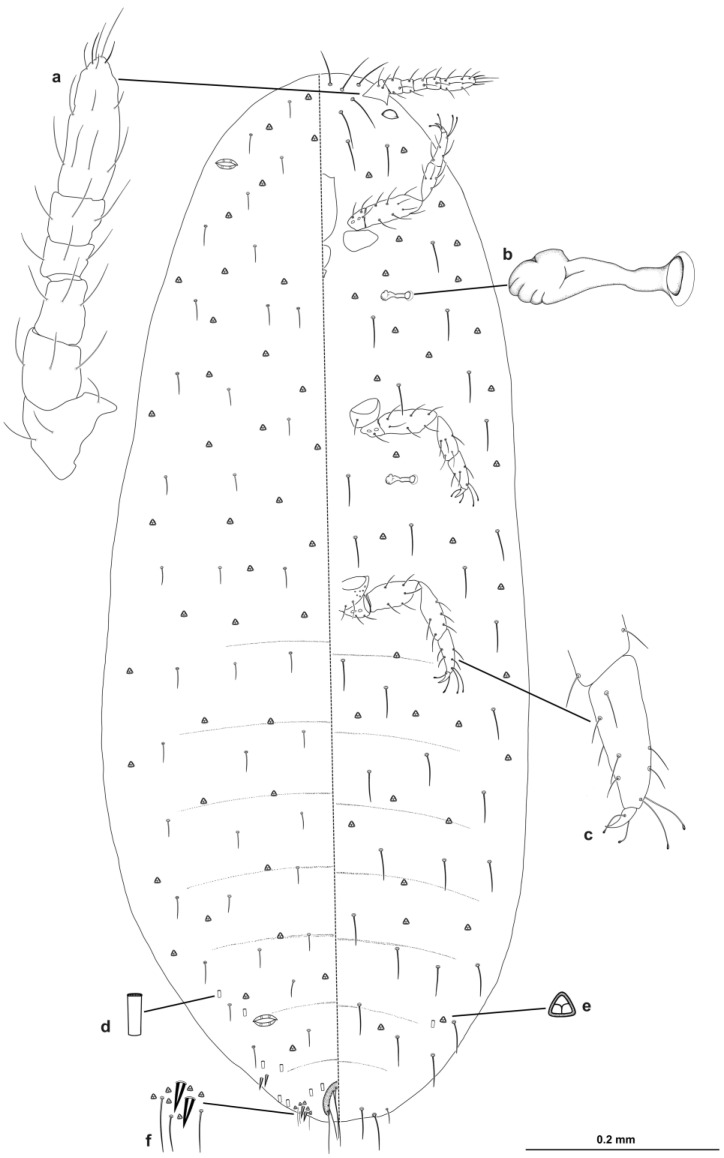
Second-instar nymph of *Trionymus aberrans* Goux 1938; (**a**) antenna; (**b**) anterior spiracle; (**c**) tarsus and claw of the hind leg; (**d**) tubular duct; (**e**) swirled trilocular pore; (**f**) anal lobe cerarii (C18).

**Table 1 insects-14-00945-t001:** Morphological features of adult females of *Trionymus aberrans* Goux, 1938.

Locality	Holotype	Site 1	Site 2	Site 3	Site 4	Site 5	Undegraded Habitats in Poland and Czech Republic	Undegraded Habitat in Austria
Morphological Feature								
Number of antennal segments (number of individuals—N)	8	8 (15)	8 (15)	8 (15)	8 (12) 9 (3)	8 (13) 9 (2)	8 (131) 9 (5)	8 (12) 9 (1)
Number of circulus (N)	0	0 (15)	0 (15)	0 (15)	0 (15)	0 (1) 1 (5)	0 (136)	0 (13)
Translucent pores on hind coxae (N)	many	a few (10) many (5)	a few (10) many (5)	a few (15)	a few (10) many (5)	a few (15)	many (136)	many (13)
Number of unusual quadrilocular pores on venter of abdomen (N)	-	0 (15)	0 (15)	0 (15)	0 (15)	0 (8) 6 (3) 7 (1) 8 (1) 10 (2)	0 (136)	0 (13)
Number of trilocular pores in C17 (N)	2	5 (15)	5 (15)	3 (12) 4 (2) 5 (1)	5 (15)	5 (15)15	6 (10) 10 (126)	5 (13)
Number of trilocular pores in C18 (N)		6 (10) 7 (2) 10 (2)	6 (13) 7 (2)	6 (11) 7 (3)	10 (15)	6 (10) 7 (5)	10 (110) 11 (5) 14 (21)	15 (6) 16 (7)
Number of conical setae in C 18 (N)		2 (15)	2 (15)	3 (4) 2 (11)	2 (15)	2 (15)	2 (136)	2 (13)

(N)—Number of individuals with particular feature; Site 1—zinc heap in Ruda Slaska; Site 2—zinc heap in Piekary Slaskie; Site 3—zinc heap in Boleslaw; Site 4—coal spoil mine heap in Ruda Slaska; Site 5—degraded habitat near zinc production plant.

## Data Availability

Data is contained within the article. The data presented in this study are available on request from the corresponding author. The data are not publicly available due to the large number of SEM photos.

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
