# Peer review of "Intraspecific Variability in Trionymus aberrans Goux, 1938 (Hemiptera: Coccomorpha: Pseudococcidae) with Description of the Second-Instar Nymph"

_insects, 2023, doi:10.3390/insects14120945_

Round 1

Reviewer 1 Report

Comments and Suggestions for Authors

Comments on: 

Morphological study of Trionymus aberrans Goux 1938 (Hemiptera: Coccoidea: Pseudococcidae) with a description of a second-instar nymph

In green my suggested changes, in yellow what I suggest deleting. In red my comments

Suggested title: Intraspecific variability in Trionymus aberrans Goux 1938 (Hemiptera: Coccoidea: Pseudococcidae) with description of a second-instar nymph

24–26 The importance of introducing additional morphological characters into the species description is discussed. New data on the frequency and host preference of T. aberrans are also provided. Our research is the first long-time study on scale insect species in post-industrial wastelands. The second-instar nymph of T. aberrans is described and illustrated. The presence of translucent pores on the hind coxae of the second-instar nymph is reported.

66 taxonomic characters that are used in the classification of mealybugs and all scale insects.

71–73: to delete. The use of the SEM is reported in Material and Methods and the comment is unnecessary.

80–-81 Translucent pores are structures that are found in many mealybug species. The function of these pores is not unclear, but they possibly release pheromones since they occuralmost solely in adult females [14] Please, move this statement after line 70.

83–86 “The aims of this study were to make the comparison of the morphology and morphological anomalies between female specimens of T. aberrans collected in post-industrial wastelands and those found in undegraded habitats using light and scanning electron microscope (SEM) and to describe the second-instar nymph of this species”.

Suggested change:

83–86 The aim of this study was to compare the morphology and to highlight possible morphological differences between females of T. aberrans collected in post-industrial wastelands and those found in undegraded habitats. Moreover, the collection of some T. aberrans second-instar nymphs gave the opportunity to describe the morphology of this stage, so far unknown.

88-91 Adult females of Trionymus aberrans were mainly collected in degraded post-industrial wastelands including three zinc spoil heaps that are located in Ruda ÅšlÄ…ska (site no. 1), Piekary ÅšlÄ…skie (no. 2) and BolesÅ‚aw (no. 3), one coal heap in Ruda ÅšlÄ…ska (no. 4) and a degraded area near a production plant in Piekary ÅšlÄ…skie  Are these localities districts or towns?

98- 100 All of the sites are located in the southern part of Poland. Specimens were also collected in undegraded habitats (unnumbered sites), primarily in psammophilous and xerothermic grasslands in Poland (also in Ruda Śląska, Piekary Śląskie and Bolesław),in the same localities reported above, plus in Austria and the Czech Republic.

103–104 The females were collected from the leaves or leaf sheaths of different grass species, using the hand-collecting method and preserved in 70% ethanol and mounted. ”: it is unclear if the specimens have been stained or not

114-118: these are “Results” not Material and Methods

119–127 Please delete, there are too many details:. For the SEM, the method of Kanturski et al. [16] was adapted and used to dehydrate and dry the specimens. The material was dehydrated in a graded ethanol/water series of 12075%, 80%, 90%, 95% and 100% for 10 min at each concentration with three 100% ethanol changes. The samples were critical point dried in a Leica EM CPD300 automated critical point dryer (Leica Microsystems, Vienna, Austria), mounted on aluminium stubs with double-sided adhesive carbon tabs and sputter coated with gold (20-30 nm thick film) in a Pelco SC-6 sputter coater (Ted Pella, Inc., Redding, CA, USA).

The images were captured using a Hitachi SU8010 field emission scanning electron Microscope. (Hitachi High-Technologies Corporation, Tokyo, Japan) at 5.0, 7.0 and 10.0 kV

accelerating voltages at the IBBEP.

129–133 The description of the second-instar nymph of T. aberrans is based on three specimens.

The description scheme was adapted from Kaydan et al. [17] and was slightly modified. The morphological terminology of the wax glands (pores and ducts) as well as the numeration of the abdominal segments followed those of Danzig and Gavrilov-Zimin [3].

Almost all of The Measurements were performed in micrometers (μm); only the body length and width were measured in millimeters (mm).

137–141 In A total of 225 female specimens of Trionymus aberrans were examined. Of these, 75 were collected in post-industrial wastelands (15 from each site) and 150 in undegraded habitats. Specimens collected in undegraded habitats including specimens collected in psammophilous grasslands located in Piekary ÅšlÄ…skie, BolesÅ‚aw and Ruda ÅšlÄ…ska. Fifteen specimens from each postindustrial wasteland were examined

144–147 A total of about 900 adult females of Trionymus aberrans were collected, most of them (620) from the post-industrial wastelands. About 40% of the collected females were parasitised. The highest degree of parasitisation was observed characteristic of the females that had been collected from the different habitats between August and October.

148 The females had been found on the leaves or in the leaf sheaths of different grass Poaceae

149 – 156 In the post-industrial wastelands, about 90% of the specimens were collected from Deschampsia caespitosa (L.) Palisot de Beauvois. Other recorded host plants in the degraded habitats grass species, e.g., were Agrostis capillaris L., Elymus repens (L.) Gould, Phleum pratense L. and Poa compressa L. were other common host. The adult females formed colonies of 4-10 consisting of four to ten specimens only on Deschampsia caespitosa in the post-industrial wastelands.

 In the undegraded habitats (e.g., psammophilous or xerothermic grasslands) present study, specimens of T. aberrans was were often collected from Brachypodium pinnatum (L.) and Phleum sp.. In the undegraded habitats . The females most often occurred singly on their host plants.

So, according your data, in post industrial waste lands the host plants were Deschampsia caespitosa Agrostis capillaris L., Elymus repens Phleum pratense L. Poa compressa whereas in undegraded habitats only Brachypodium pinnatum (L.) and Phleum

159 –250 Intraspecific variability in adult females of Trionymus aberrans

As I understood this is a detailed redescription of Trionymus aberrans based on the morphology of type specimen, 5 specimens from MNHN and your 225 specimens from post-industrial degraded areas and undegraded areas.

160 –171     225 adult female specimens were involved in the morphological examination, out of the ca. 900 T. aberrans adult females that were collected during the present studies and deposited in the entomological collections of the Zoology Research Group, Institute of Biology, Biotechnology and Environmental Protection, Faculty of Life Sciences, University of Silesia, Poland (supplementary material, Table S1)

Comment: you have already reported (see line 135) that he specimens of T. aberrans collected during the present studies are deposited in the IBBEP.  May I suggest you to add the whole name of the Institution  at line 135 plus the acronym in brackets  and  here just insert the acronym (see below)

Material examined:

Trionymus aberrans n. sp. / Hem. Coccidae / ♀ / Avena / Marseille / 1.VIII.1936 / 806 /a / 162

Type / Holotype (MNHN)

Trionymus aberrans ovalis/ Type / Colmars (Basses-Alpes) / L. Goux coll / 19/VII/1939/ 164

1013 (MNHN)

3 adult females deposited in MNHN

225 adult females collected during the present study deposited in IBBEP

179  Description of a slide-mounted adult female – Body elongate oval, 4.25–4.92 mm long…etc.,

From 179 to 241 It is a redescription of the species, based on all the studied specimens. Very long and detailed and not necessary as the species was already descripted by different authors.

In your abstract you report “A distinctive morphological variability was observed in the specimens that had been collected from post-industrial wastelands in Poland.

In fact, in a so long and detailed description is quite difficult to find out and compare data from the different collecting sites and this doesn’t help in understanding the morphological variability in specimens from degraded and undegraded sites.

I suggest you focusing on the “distinctive variability” and this is not possible if you present your data all together and not separate in “specimens from degraded habitats” and “specimens from undegraded habitats”

What I strongly suggest (and in my opinion it is mandatory) is: all the data on numerical morphological differences among specimens from the different collecting sites must be inserted in a table so that one can clearly understand what the differences are.

The table must report in the columns the observed sites, plus holotype: 1) Holotype 2) each of the 5 degraded sites 3) each of the undegraded sites, and in the rows the numerical morphological data/site: nr antennomera, nr. translucent pores, presence/absence of circulus, number of trilocular pores in C17, C18, unusual quadrilocular pores  etc..

This way the reader can understand the intraspecific variability of the species and the possible differences among specimens from the different sites. This suggestion was already reported in the first review.

The title of this part is the one reported at line 159: Intraspecific variability in adult females

248– 252 Morphological anomalies were observed in females collected in four of the examined post-industrial wastelands, namely zinc heaps (site no. 1, 2 and 3) and degraded area near the production plant (site no. 5). Which are these morphological anomalies? “Specimens collected in coal spoil heap (site no. 4) possess only fewer number of translucent pores in hind coxae compared to specimens found undegraded habitats” : this is intraspecific variability

253-263 Unusual pores were observed in seven females out of fifteen examined specimens that had been collected from the site no.5, one of the post-industrial wastelands namely, the degraded area near the production plant in Piekary ÅšlÄ…skie.

The females with unusual pores were collected in 2017, 2018 and 2020. The detailed structure of these unusual pores were accurately observed using a scanning electron microscope (SEM). These pores were characterised by the presence These unusual pores were scattered on the ventral surface of the last three abdominal segments how many unusual pores/specimen?. They were somewhat similar in appearance to the usual trilocular pores but had one more excretory opening so being quadriloculars of four excretory openings (loculi) (Fig. x ).

No pores with four loculi were observed in females that had been collected from the undegraded habitats nor in the females that had been deposited in from the collections in MNHN and IBBEP.

Fig. 2 Figure 2. Adult females of Trionymus aberrans: delete as unnecessary, nothing new to observe

Fig. Fig. 4: please, substitute the fig.4 c (simple tubular duct) with Fig. 3 c (trilocular pore), for a better comparison between the new quadrilocular and the usual trilocular pore. Eliminate the   fig. 3, as unnecessary. Please change the caption o. 

269 – 308 This part refers on presence or absence of circulus

Five adult females out of fifteen examined specimens that had been collected in the degraded area near a production plant in Piekary ÅšlÄ…skie (site no. 5) had a small and oval circulus between the third and fourth abdominal segments (Fig. 2 (b-d)). The females with circulus were collected in 2015, 2017 and 2020. The circulus was observed in a type specimen of Trionymus aberrans ovalis. The circulus was absent in all the females that been collected from the undegraded habitats and in almost all what means “almost all? of the females deposited the examined collections in MNHN and IBBEP. The circulus was also absent in the vast majority? of specimens that had been collected from post-industrial habitats.

What can you infer about these data?  Simply that circulus is usually absent and sometimes present, as already reported by Goux, Gavrilov and other authors.

309- 310 In specimens collected in undegraded psammophilous grasslands in Ruda ÅšlÄ…ska,

Piekary Śląskie and Bolesław morphological anomalies? were not observed.

314 Description of a second-instar nymph of Trionymus aberrans

316 Trionymus aberrans / ♀ / larva / Graminae vagina ???/ det. Koteja // 5.9.1967 / 6

318-321 Trionymus aberrans / nymph / Phleum sp. / Olsztyn near Częstochowa / 12.09.2018 // leg. et det. M. Kalandyk-Kołodziejczyk;

Trionymus aberrans / nymph / Deschampsia caespitosa / Piekary ÅšlÄ…skie, Lotników (site n. 5?)/

12.09.2020 // leg. et det. M. Kalandyk-Kołodziejczyk

377 -383  4. Discussion

Although studies on the occurrence of scale insects in anthropogenic habitats have been conducted in Poland and other countries [21, 22, 23], these studies did not include the coccoid fauna from post-industrial wastelands. We are not aware of any research that has been conducted

Very few researches have been conducted on the scale insects from post-industrial wastelands other than those of Lubiarz and Golan [24], Lubiarz and Cichocka [25], Kalandyk-KoÅ‚odziejczyk et al. [26] and Mróz et al. [13]. Results of these studies?? Not reported

Our study is the first long-time study conducted on scale insect species found in post-industrial wastelands.

385 – 390 “Differences in the abundance and host-plant preferences have been observed between the adult females of Trionymus aberrans from the post-industrial wastelands and those living in undegraded habitats in Poland and other countries. In habitats such as xerothermic and psammophilous grasslands, one or two adult females have usually been observed on a single grass [27, 28], while in the degraded areas that were examined in this study, from four to ten females were found on a single host plant.” According to your data (149-156) you collected T. aberrans from sites and from different host plants so that data are not comparable. Moreover 4-10 females /plant were collected only on Deschampsia cespitosa, a perennial plant where a scale insect population can establish and persist.

404 – 405 Our research indicates that T. aberrans has different trophic preferences in degraded habitats than in other habitats. Which differences?  Based on?? Was the flora of the degraded habitats different from the undegraded ones?  If so, there is no possibility of choice or “preference”.

489 Supplementary Materials: Table S1: List of examined microscope slides of adult females of Trionymus aberrans Goux, 1938:  not necessary.

FIGURES

Fig. 1: Map not necessary

Fig. 2 : Adult females of Trionymus aberrans: unnecessary, nothing new to observe

Fig. 4: please, substitute the fig.4 c (simple tubular duct) with Fig. 3 c (trilocular pore), for a better comparison between the new quadrilocular pore and the usual trilocular pore (this is an interesting date).

Eliminate the   fig. 3a,b,d  as unnecessary. Please, change the caption.

VWhat is worth to be published are: the discovery of a new morphological character in Pseudococcidae (presence of quadrilocular pores in a very few specimens collected in post-industrial wastelands) and the description of the second instar nymph, so far undescribed.

Regarding the interspecific variability, noticed by the authors among the many studied adult females Trionymus aberrans, in my first revision I had already strongly recommended the authors reporting their data in a table to facilitate the comparison and highlighting the possible differences (if any) between specimens collected from degraded post-industrial wasteland and specimens collected from undegraded lands. Apparently, this was not done.

Fig. 1: Map not necessary

Fig. 2: Adult females of Trionymus aberrans: unnecessary, nothing new to observe

Figg. 3-4:  to gather in only one Fig. by eliminating some photos (see in comments for Authors)

What is worth to be published are: the discovery of a new morphological character in Pseudococcidae (presence of quadrilocular pores in a very few specimens collected in post-industrial wastelands) and the description of the second instar nymph, so far undescribed.

Regarding the interspecific variability, noticed by the authors among the many studied adult females Trionymus aberrans, in my first revision I had already strongly recommended the authors reporting their data in a table to facilitate the comparison and highlighting the possible differences (if any) between specimens collected from degraded post-industrial wasteland and specimens collected from undegraded lands. Apparently, this was not done.

Fig. 1: Map not necessary

Fig. 2: Adult females of Trionymus aberrans: unnecessary, nothing new to observe

Figg. 3-4:  to gather in only one Fig. by eliminating some photos (see in comments for Authors)

Author Response

I Responses to the first reviewer’s comments

We would like to thank for the insightful review of our manuscript.

We have numbered the comments and our answers are underlined.

The sentences in yellow have been removed and those in green have been changed.

  1. Suggested title: Intraspecific variability in Trionymus aberrans Goux 1938 (Hemiptera: Coccoidea: Pseudococcidae) with description of a second-instar nymph.

1.The title of the manuscript has been changed according to the Reviewer’s suggestion.

  1. 24–26 The importance of introducing additional morphological characters into the species description is discussed. New data on the frequency and host preference of T. aberrans are also provided. Our research is the first long-time study on scale insect species in post-industrial wastelands. The second-instar nymph of T. aberrans is described and illustrated. The presence of translucent pores on the hind coxae of the second-instar nymph is reported.
  2. The changes have been implemented.
  3. 66 taxonomic characters that are used in the classification of mealybugs andall scale insects.
  4. 71­–73: to delete. The use of the SEM is reported in Material and Methods and the comment is unnecessary.

3 and 4. Sentences have been removed.

  1. Translucent pores are structures that are found in many mealybug species. The function of these pores is notunclear, but they possibly release pheromones since they occuralmost solely in adult females [14] Please, move this statement after line 70.
  2. This statement has been moved after line 70.
  3. 83–86 “The aims of this study were to make the comparison of the morphology and morphological anomalies between female specimens of T. aberrans collected in post-industrial wastelands and those found in undegraded habitats using light and scanning electron microscope (SEM) and to describe the second-instar nymph of this species.”

Suggested changes:

The aim of this study was to compare the morphology and to highlight possible morphological differences between females of T. aberrans collected in post-industrial wastelands and those found in undegraded habitats. Moreover, the collection of some T. aberrans second-instar nymphs gave the opportunity to describe the morphology of this stage, so far unknown.

  1. The changes have been implemented.
  2. 88-91 Adult females of Trionymus aberrans were mainly collected in degraded post-industrial wastelands including three zinc spoil heaps that are located in Ruda ÅšlÄ…ska (site no. 1), Piekary ÅšlÄ…skie (no. 2) and BolesÅ‚aw (no. 3), one coal heap in Ruda ÅšlÄ…ska (no. 4) and a degraded area near a production plant in Piekary ÅšlÄ…skie  Are these localities districts or towns?
  3. Information about localities has been completed.
  4. 98- 100 All ofthe sites are locatedin the southern part of Poland. Specimens were also collected in undegraded habitats (unnumbered sites), primarily in psammophilous and xerothermic grasslands in Poland (also in Ruda ÅšlÄ…ska, Piekary ÅšlÄ…skie and BolesÅ‚aw),in the same localities reported above, plus in Austria and the Czech Republic.
  5. The changes have been implemented.
  6. 103–104 The females were collected from the leaves or leaf sheaths of different grass species, using the hand-collecting method and preserved in 70% ethanol and mounted. ”: it is unclear if the specimens have been stained or not
  7. Information about staining has been completed.
  8. 114-118: these are “Results” not Material and Methods.
  9. These sentences have been moved to “Results”.
  10. 119–127 Please delete, there are too many details:. For the SEM, the method of Kanturski et al. [16] was adapted and used to dehydrate and dry the specimens. The material was dehydrated in a graded ethanol/water series of 12075%, 80%, 90%, 95% and 100% for 10 min at each concentration with three 100% ethanol changes.The samples were critical point dried in a Leica EM CPD300 automated critical point dryer (Leica Microsystems, Vienna, Austria), mounted on aluminium stubs with double-sided adhesive carbon tabs and sputter coated with gold (20-30 nm thick film) in a Pelco SC-6 sputter coater (Ted Pella, Inc., Redding, CA, USA)

images were captured using a Hitachi SU8010 field emission scanning electron Microscope. (Hitachi High-Technologies Corporation, Tokyo, Japan) at 5.0, 7.0 and 10.0 kV

accelerating voltages at the IBBEP

  1. Sentences and words in yellow have been deleted.
  2. 129–133 The description of the second-instar nymph of T. aberrans is based on three specimens

description scheme was adapted from Kaydan et al. [17] and was slightly modified. The morphological terminology of the wax glands (pores and ducts) as well as the numeration of the abdominal segments followed those of Danzig and Gavrilov-Zimin [3].

Almosta all of The Measurements were performed in micrometers (μm); only the body length and width were measured in millimeters (mm).

  1. Sentences and words in yellow have been removed.
  2. 137–141 InA total of225 female specimens of Trionymus aberrans were examined. Of these, 75 were collected in post-industrial wastelands (15 from each site) and 150 in undegraded habitats. Specimens collected in undegraded habitats including specimens collected in psammophilous grasslands located in Piekary ÅšlÄ…skie, BolesÅ‚aw and Ruda ÅšlÄ…ska. Fifteen specimens from each postindustrial wasteland were examined.
  3. The changes have been implemented.
  4. 144–147 A total of about 900 adult females of Trionymus aberrans were collected, most of them (620) from the post-industrial wastelands. About 40% of the collectedfemales were parasitised. The highest degree of parasitisation was observedcharacteristic of the females that had been collected from the different habitats between August and October.
  5. 148 The females had been found on the leaves or in the leaf sheaths of different grassPoaceae.

14 and 15. The changes have been implemented.

  1. 149–156 In the post-industrial wastelands, about 90% of the specimens were collected from Deschampsia caespitosa (L.) Palisot de Beauvois. Other recorded host plants in the degraded habitats grass species, e.g.,were Agrostis capillaris L., Elymus repens (L.) Gould, Phleum pratense L. and Poa compressa L. wereother common host. The adult females formed colonies of 4-10 consisting of four to ten specimens only on Deschampsia caespitosa in the post-industrial wastelands.

In the undegraded habitats (e.g., psammophilous or xerothermic grasslands) present study, specimens of T. aberrans was were often collected from Brachypodium pinnatum (L.) and Phleum sp.. In the undegraded habitats . The females most often occurred singly on their host plants.

So, according your data, in post industrial waste lands the host plants were Deschampsia caespitosa Agrostis capillaris L., Elymus repens Phleum pratense L. Poa compressa whereas in undegraded habitats only Brachypodium pinnatum (L.) and Phleum.

  1. The changes suggested by the Reviewer have been implemented. The words in yellow have been deleted.

We did not present the observations in sufficient details in our manuscript. We had added necessary information to the “Discussion”.

Deschampsia caespitosa and Brachypodium pinnatum were common species in many sites including both post-industrial wastelands and undegraded habitats. Despite this fact, in post-industrial areas females of Trionymus aberrans occurred mostly on Deschampsia caespitosa and only a small percentage of specimens were found on different grasses, but not on Brachypodium pinnatum, which was common in these habitats. In several undegraded habitats Deschampsia caespitosa was a dominant grass species, but in these sites Trionymus aberrans was not found on this grass species. In undegraded habitats some other scale insects species were collected from D. caespitosa, but any specimens of T. aberrans.

  1. 159–250 Intraspecific variability in adult females of Trionymus aberrans.

I understood this is a detailed redescription of Trionymus aberrans based on the morphology of type specimen, 5 specimens from MNHN and your 225 specimens from post-industrial degraded areas and undegraded areas.

  1. Our redescription of Trionymus aberrans is based on the 5 specimens from MNHN (including types) and 225 specimens from post-industrial wastelands and undegraded areas. We added information to text.
  2. 225 adult female specimens were involved in the morphological examination, out of the ca. 900 T. aberrans adult females that were collected during the present studies and deposited in the entomological collections of the Zoology Research Group, Institute of Biology, Biotechnology and Environmental Protection, Faculty of Life Sciences, University of Silesia, Poland (supplementary material, Table S1.

Comment: you have already reported (see line 135) that he specimens of T. aberrans collected during the present studies are deposited in the IBBEP.  May I suggest you to add the whole name of the Institution  at line 135 plus the acronym in brackets  and  here just insert the acronym (see below).

  1. The words in yellow have been removed. The whole name of the Institution has been placed elsewhere in the text.
  2. Material examined

Trionymus aberrans n. sp. / Hem. Coccidae / ♀ / Avena / Marseille / 1.VIII.1936 / 806 /a / 162

Type/Holotype (MNHN)

Trionymus aberrans ovalis/ Type / Colmars (Basses-Alpes) / L. Goux coll / 19/VII/1939/ 16

1013 (MNHN)

3 adult females deposited in MNHN

225 females collected during the present study deposited in IBBEP.

  1. The sentence has been added to the text.

Description of a slide-mounted adult female – Body elongate oval, 4.25–4.92 mm long…etc.

  1. From 179 to 241 It is a redescription of the species, based on all the studied specimens. Very long and detailed and not necessary as the species was already descripted by different authors.

In fact, in a so long and detailed description is quite difficult to find out and compare data from the different collecting sites and this doesn’t help in understanding the morphological variability in specimens from degraded and undegraded sites.

I suggest you focusing on the “distinctive variability” and this is not possible if you present your data all together and not separate in “specimens from degraded habitats” and “specimens from undegraded habitats.

I strongly suggest (and in my opinion it is mandatory) is: all the data on numerical morphological differences among specimens from the different collecting sites must be inserted in a table so that one can clearly understand what the differences are.

The table must report in the columns the observed sites, plus holotype: 1) Holotype 2) each of the 5 degraded sites 3) each of the undegraded sites, and in the rows the numerical morphological data/site: nr antennomera, nr. translucent pores, presence/absence of circulus, number of trilocular pores in C17, C18, unusual quadrilocular pores etc.

  1. According to the suggestions, we have presented the results in the table with columns titled: holotype, each of 5 degraded sites, undegraded sites in Poland and the Czech Republic and separately undegraded habitat in Austria. Undegraded habitat in Austria was separated because the specimens collected in this site possessed larger number of trilocular pores in C18 than those from other undegraded habitats.

In rows there are morphological data.

Each column is subdivided int two subcolumns, the left includes number of structure e.g. number of antennal segment, the right contains number of specimen.

The table could not contain more columns due to its size restrictions.

We have removed from the text all sentences whose content is presented in the table.

We would like to maintain the redescription of Trionymus aberrans, because we think that it can be used for further research on the intraspecific variability of this species.

  1. Morphological anomalies were observed in females collected in four of the examined post-industrial wastelands, namely zinc heaps (site no. 1, 2 and 3) and degraded area near the production plant (site no. 5). Which are these morphological anomalies? “Specimens collected in coal spoil heap (site no. 4) possess only fewer number of translucent pores in hind coxae compared to specimens found undegraded habitats” : this is intraspecific variability
  2. Information has been added to the text.
  3. Unusual pores were observed in seven females out of fifteen examined specimens that had been collected from the site no.5, one ofthe post-industrial wastelands namely, the degraded area near the production plantin Piekary ÅšlÄ…skie

The females with unusual pores were collected in 2017, 2018 and 2020. The detailed structure of these unusual pores were accurately observed using a scanning electron microscope (SEM). These pores were characterised by the presence These unusual pores were scattered on the ventral surface of the last three abdominal segments how many unusual pores/specimen?. They were somewhat similar in appearance to the usual trilocular pores but had one more excretory opening so being quadriloculars of four excretory opening (loculi) (Fig.x).

  1. The words in yellow have been removed. Information about number of pores per specimen has been added to the text.
  2. 2 Figure 2. Adult females of Trionymus aberrans: delete as unnecessary, nothing new to observe.
  3. The figure has been removed.
  4. Fig. Fig. 4: please, substitute the fig.4 c (simple tubular duct) with Fig. 3 c (trilocular pore), for a better comparison between the new quadrilocular and the usual trilocular pore. Eliminate the   fig. 3, as unnecessary. Please change the caption o
  5. All proposed changes have been applied. There is now one figure 2 with quadrilocular pore and trilocular pore. Captions have been changed.
  6. 269– 308 This part refers on presence or absence of circulus. Five adult females out of fifteen examined specimens that had been collected in the degraded area near a production plant in Piekary ÅšlÄ…skie (site no. 5) had a small and oval circulus between the third and fourth abdominal segments (Fig. 2 (b-d)). The females with circulus were collected in 2015, 2017 and 2020. The circulus was observed in a type specimen of Trionymus aberrans ovalis. The circulus was absent in all the females that been collected from the undegraded habitats and in almost all what means “almost all? of the females deposited the examined collections in MNHN and IBBEP. The circulus was also absent in the vast majority?of specimens that had been collected from post-industrial habitats.

What can you infer about these data?  Simply that circulus is usually absent and sometimes present, as already reported by Goux, Gavrilov and other authors

  1. We have removed all sentences regarding presence or absence of circulus except for the only one: “The small circulus was rarely observed in females collected during present studies”.
  2. Description of a second-instar nymph of Trionymus aberrans

Trionymus aberrans / ♀ / larva / Graminae vagina ???/ det. Koteja // 5.9.1967 / 6

  1. We think, that the correct form is “leaf sheaths of Poaceae”, but we copied information from original label.
  2. Trionymus aberrans/ nymph / Deschampsia caespitosa/ Piekary ÅšlÄ…skie, Lotników (site n. 5?)/
  3. These words have been added.
  4. Although studies on the occurrence of scale insects in anthropogenic habitats have been conducted in Poland and other countries [21, 22, 23], these studies did not include the coccoid fauna from post-industrial wastelands.We are not aware of any research that has been conducted.

Very few researches have been conducted on the scale insects from post-industrial wastelands other than those of Lubiarz and Golan [24], Lubiarz and Cichocka [25], Kalandyk-KoÅ‚odziejczyk et al. [26] and Mróz et al. [13]. Results of these studies?? Not reported

  1. We have removed sentences in yellow and changed this part according to the suggestions. We have added information about results of these studies. These references are also cited below in the Discussion.
  2. 385-390 Differences in the abundance and host-plant preferences have been observed between the adult females of Trionymus aberrans from the post-industrial wastelands and those living in undegraded habitats in Poland and other countries. In habitats such as xerothermic and psammophilous grasslands, one or two adult females have usually been observed on a single grass [27, 28], while in the degraded areas that were examined in this study, from four to ten females were found on a single host plant.” According to your data (149-156) you collected T. aberrans from sites and from different host plants so that data are not comparable. Moreover 4-10 females /plant were collected only on Deschampsia cespitosa, a perennial plant where a scale insect population can establish and persist.

404-405 Our research indicates that T. aberrans has different trophic preferences in degraded habitats than in other habitats. Which differences?  Based on?? Was the flora of the degraded habitats different from the undegraded ones?  If so, there is no possibility of choice or “preference

  1. In the manuscript we did not include necessary information about plant species in post-industrial wastelands and undegraded habitats. Deschampsia caespitosa was a common species in all wastelands and many undegraded areas, but we found females of Trionymus aberrans on this grass only in post-industrial habitats. On the other hand, Brachypodium pinnatum was present in two types of habitats, but T. aberrans were found on this grass only in undegraded habitats. Taking into account the presence of the same plant species in two types of habitats (degraded and undegraded), we can assume that the trophic preferences of adult females of T. aberrans differ in different habitats. We have added an explanation to the Discussion.
  2. Supplementary Materials: Table S1: List of examined microscope slides of adult females of Trionymus aberransGoux, 1938:  not necessary
  3. We would like to leave a list of slides because they can be borrowed by other researchers.
  4. Fig 1: Map not necessary.
  5. We would decide to leave the map as an illustration of the location of the sites in the field and the distance between them.

Reviewer 2 Report

Comments and Suggestions for Authors

The manuscript covers a topic that has been little researched and the new information that it provides is potentially of considerable significance. However, in the Introduction, some information from ScaleNet needs updating (please see my suggestions in Track Changes in the Word file) and 2 more references need to be cited. The Results need to be presented more clearly - addition of some tables would make the data more accessible at a glance, rather than trying to present all the data in sentences. You need some sort of measure of the relative levels of zinc contamination at sites 1-3 and 5, before you can make informed deductions about why the specimens from site 5 differ so markedly from those from the other 3 zinc-contaminated sites. If the morphological changes observed can be related to a higher level of zinc contamination, there is a potential for them to be used as indicators of environmental quality. The discussions and conclusions are overgeneralised in places. They need some more analytical thought; you should be able to draw more specific deductions from the information you have gathered. Two more references need to be added to the References section. In the Supplementary material, Table 1, the column headings can be improved. I can make suggestions on that, but this online system does not allow me to upload more than one edited file.

Comments on the Quality of English Language

The English needs improvement throughout; I have made suggestions and precis in Track Changes throughout the Word copy attached.

Author Response

We would like to thank for all suggestions and English improvement.

We have introduced suggested changes and presented the results in the form of table. We have made required additions to the text.

We added one proposed reference: Choi and Lee 2022. We could not find information about the second reference.

We have also added publications containing information about zinc and other heavy metals in the examined sites.

The paper has also been changed according to other reviewers’ remarks.

Reviewer 3 Report

Comments and Suggestions for Authors

This manuscript presents novel findings on a morphological variation of the adult females of Trionymus aberrans, along with the description of its second-instal nymph. In addition, the authors compared the morphology of T. aberrans sampled from degraded/undegraded habitats. Despite the authors’ commendable efforts, the manuscript faces several critical issues. The key concerns are outlined below:

1. Unclear definition of sampling sites

This study aims to compare the morphology of T. aberrans from different environmental conditions, such as degraded post-industrial wastelands and undegraded habitats. However, the regions of interest lack clear definitions. What are the detailed differences between post-industrial wastelands and undegraded habitats? It is also unclear that sampling sites of post-industrial wastelands are the regions with the same environmental conditions. Please see Figure 1. It indicates the five sampling sites of post-industrial wastelands, but they are placed in different environmental categories, such as urban areas and agricultural areas. Moreover, heterogeneous patches with different environmental conditions can exist even in the same place of wastelands. Therefore, the authors need to scientifically define and differentiate sampling sites.

2. Uncertain reason for morphological comparison

The authors compared the morphological differences of samples from degraded/undegraded habitats. However, the paper lacks a background or hypothesis regarding the relationship between environmental conditions and the morphological variation of mealybugs. There should be various biological/environmental reasons to affect the wax-secreting organs like pores and ducts of mealybugs. Which environmental condition can specifically affect the morphological change/variation of the mealybugs? Especially, how can link this morphological variation to poorly defined environmental conditions in this study? Moreover, the samples were collected from several different host plants, which means other environmental conditions were not well controlled. Therefore, the authors should address the clear reasons for this comparison and control the other potential factors affecting the morphological variation. Furthermore, this kind of the studies requires better sampling strategies to statistically validate the results of the research. 

3. Incomplete formatting of morphological description and other contents

The authors partitioned the description with “Instar diagnosis”, “Description of a slide-mounted adult female”, “Venter” and “Dorsum”. Although there are sessions for morphological characters on the dorsum and venter, several dorsal and ventral characters (e.g. antennae, clypeolabral shield, spiracles, legs, etc.) are described out of the appropriate sessions like “Venter” and “Dorsum”. It is also not understandable the presence of “instar diagnosis” in the description for the adult female because “the instar” indicates the developmental stages of nymphs. The host plants of the samples are also missing in the supplementary table. Although the authors described morphological variations among samples in the comment session, it is not easy to recognize the morphological differences because of complex explanations including multiple sampling sites. Moreover, many contents are repeated in the discussion. Therefore, the description of the morphological differences should be summarized and presented in a table, which allows readers to easily understand the morphological variation of T. aberrans according to the habitats.  

4. Drawback of the description of the second-instar

The authors should address how they can identify the examined samples in the second-instar instead of the first and third instars. In addition, it should be compared to the morphology of the different species in the same instar, if the descriptions are available. The format of the description has the same problem as the adult females (descriptions of dorsal and ventral characters are placed out of “Venter” and “Dorsum” sessions).

In conclusion, I recommend rejecting this paper for publication in Insects due to the aforementioned issues. A resubmission should involve reformatting the manuscript, providing a more concise description of T. aberrans, showcasing its morphological variation, and addressing the concerns related to the second-instar description.

Comments on the Quality of English Language

Many parts of the manuscript are too wordy. Many sentences should be summarized and rephrased.

Author Response

We would like to thank you for your comments and suggestions and we informed you that this paper has been changed also according to other reviewers’ remarks.

  1. Unclear definition of sampling sites

The studied sites are not distinguished on the map used due to their small areas. All of these sites are industrially contaminated with heavy metals. All examined areas have been tested for heavy metal content and they differ significantly from neighboring areas. We added publications to the References containing information about these contaminations (16-19).

  1. Uncertain reason for morphological comparison

Studies on the morphological intraspecific in mealybugs and other scale insect species in post-industrial wasteland have not been conducted so far.

Other authors indicated that the number and distribution of cuticular structures e.g. pores and ducts are caused by the influence of environmental factors (these results are reported in publications from the references: 25, 32, 34 and 35).

Further studied will be conducted on morphology of mealybugs from post-industrial contaminated areas.

  1. Incomplete formatting of morphological description and other contents

We have added table to present our results.

The descriptions of the adult female and second-instar nymph is presented in accordance with the convention adopted in this group of insects.

The body shape, antennae, legs, anal ring, spiracles are described separately. In the parts “Venter” and “Dorsum” only cuticular structures are described e.g. pores and ducts, their shape, number and distribution.

We have replaced the term “instar” with “diagnosis of adult female” and “diagnosis of second-instar nymph”.

  1. Drawback of the description of the second-instar

The nymphs of different species of Trionymus are not described.

The first instar nymph  in mealybugs is characterized by the small size and presence of the 5-segmented antennae. This instar do not possess multilocular pores. The second instar is considerably larger and the ratio of the legs to the body is different. The second instar nymph has 6-segmented antennae. The multilocular pores are absent in this stage. The third instar has 7 or 8-segmented antennae and possess multilocular pores.

Round 2

Reviewer 1 Report

Comments and Suggestions for Authors

line 239 and following: please, delete  "morphological anomalies"  and change the sentence to  to:" difference in number of trilocular pores in C18  were observed in .......".

line 234:  please, delete "morphological anomalies" 

Author Response

We would like to thank you for all your comments thanks to which our manuscript has been improved.

The sentence has been changed into: Differences in number of trilocular pores in C18 were observed in females collected from the four zinc-contaminated post-industrial wasteland sample sites: zinc heaps (sites 1,2, and 3) and the degraded area near the zinc production plant at Piekary ÅšlÄ…skie (site 5).

“Morphological anomalies” has been changed into “morphological differences”.

Reviewer 2 Report

Comments and Suggestions for Authors

The revised manuscript is much improved but still needs a little attention (see my suggestions in the PDF file). The main thing that needs significant improvement is the presentation of Table 1, whose layout is not adequately explained.

Comments on the Quality of English Language

The English is much better. I made a few suggestions for improvements in the PDF file. The punctuation still needs attention (see the PDF file).

Author Response

We would like to thank you for all your comments thanks to which our manuscript has been improved.

Thank you very much for improving our English.

All suggested changes have been implemented.

According to the suggestions, we have presented the results in the table with columns titled: holotype, each of 5 degraded sites, undegraded sites in Poland and the Czech Republic and separately undegraded habitat in Austria. The undegraded habitat in Austria was separated because the specimens collected in this site possessed a larger number of trilocular pores in C18 than those from other undegraded habitats.

The table could not contain more columns due to its size restrictions.

In the columns, we have included the number of individuals with a given feature in brackets.

The table has been changed according to the reviewers’ suggestions.

We conducted a minor revision of our manuscript according to all reviewers’ suggestions.

Reviewer 3 Report

Comments and Suggestions for Authors

Thanks to the authors, I found much improvement in the revision compared to the first version of the manuscript. However, there are still many issues with text formatting/editing. Please consider the following issues and find detailed comments in the attached file.

Main issues

1. Too much-partitioned structure of the paragraphs

The sentences for explaining a certain point of the study should be included in a single paragraph. Then, each paragraph will give a clear message to the readers. However, the current version of the manuscript includes lots of sentences losing their proper paragraphs. Many sentences can be combined into the same paragraph. I made comments for many of them. Please consider them as well as other parts I didn't point out.

2. Many duplicated explanations in the comments of the description and discussion

I found many overlaps in the contents of the manuscript. Please drop some of them referring to the comments I made. In the discussion, it should be better to focus on certain main points from this study. I'd recommend dropping some content that is fully supported in this study and mostly based on the assumption. 

3. Some wrong sentence edit

I don't know the status of the original version of the authors, but I found many errors in the text edit in the revision. For example, no space between the words, no period, double commas, etc. Please carefully check and revise those errors in the manuscript.

For other issues, please find them in the attached.

Comments on the Quality of English Language

improved 

Author Response

We would like to thank you for all your comments thanks to which our manuscript has been improved.

Suggested changes have been implemented. We conducted a minor revision of our manuscript according to all reviewers’ suggestions.

We would like to leave some sentences that were suggested to be removed because in our opinion they are helpful in maintaining the sequence of information.